# Towards Universal Certified Robustness with Multi-Norm Training

## Abstract

Existing certified training methods can only train models to be robust against a certain perturbation type (e.g. $l_\infty$ or $l_2$). However, an $l_\infty$ certifiably robust model may not be certifiably robust against $l_2$ perturbation (and vice versa) and also has low robustness against other perturbations (e.g. geometric transformation). To this end, we propose the first multi-norm certified training framework **CURE**, consisting of a new $l_2$ deterministic certified training defense and several multi-norm certified training methods, to attain better *union robustness* when training from scratch or fine-tuning a pre-trained certified model. Further, we devise bound alignment and connect natural training with certified training for better union robustness. Compared with SOTA certified training, **CURE** improves union robustness up to $22.8\%$ on MNIST, $23.9\%$ on CIFAR-10, and $8.0\%$ on TinyImagenet. Further, it leads to better generalization on a diverse set of challenging unseen geometric perturbations, up to $6.8\%$ on CIFAR-10. Overall, our contributions pave a path towards *universal certified robustness*.

## 1 Introduction

Though deep neural networks (DNNs) are widely deployed in various vision applications, they remain vulnerable to adversarial attacks (Goodfellow et al., 2014; Kurakin et al., 2018). Though many empirical defenses (Madry et al., 2017; Zhang et al., 2019a; Wang et al., 2023) against adversarial attacks have been proposed, they do not provide provable guarantees and remain vulnerable to stronger attacks. Therefore, it is important to train DNNs to be *formally* robust against adversarial perturbations. Various works (Mirman et al., 2018; Gowal et al., 2018; Zhang et al., 2019b; Balunović & Vechev, 2020; Shi et al., 2021; Müller et al., 2022a; Hu et al., 2023; 2024; Mao et al., 2024) on deterministic certified training against $l_\infty$ and $l_2$ perturbations have been proposed. However, those defenses are mostly limited to a certain type of perturbation and cannot easily be generalized to other perturbation types (Yang et al., 2022). Certified robustness against multiple perturbation types is essential because this better reflects real-world scenarios where adversaries can use multiple $l_p$ perturbations. Also, Mangal et al. (2023) argue that $l_p$ robustness is the bedrock for non-$l_p$ robustness. In this work, we also show that training with multi-norm robustness can lead to stronger *universal certified robustness* by generalizing better to other perturbation types such as geometric transformations (Section 5.1).

To this end, we propose the first multi-norm **C**ertified training for **U**nion **R**obustn**E**ss (**CURE**) framework, consisting of a new $l_2$ deterministic certified training defense and several multi-norm certified training methods. Inspired by SABR (Müller et al., 2022a), our $l_2$ defense first finds the $l_2$ adversarial examples in a slightly truncated $l_2$ region and then propagates the smaller $l_\infty$ box using the IBP loss (Gowal et al., 2018). For multi-norm certified training, we propose several methods based on multi-norm empirical defenses (Tramer & Boneh, 2019; Madaan et al., 2021; Croce & Hein, 2022). Our proposed methods successfully improve both the union and universal certified robustness, as illustrated in Table 1, 2, and 2.

However, the aforementioned methods achieve sub-optimal union robustness since they do not exploit the connections between certified training for different $l_p$ perturbations as well as natural training. In Figure 1a, we show that an $l_\infty$ certified robust model may lack $l_2$ certified robustness and vice versa: $l_\infty$ model only has $5.4\%$ $l_2$ robustness and $l_2$ model has $0\%$ $l_\infty$ robustness. Thus, mitigating the tradeoff between $l_2$ and $l_\infty$ certified training is crucial. For given values of $\epsilon_q$ and $\epsilon_r$, we compare

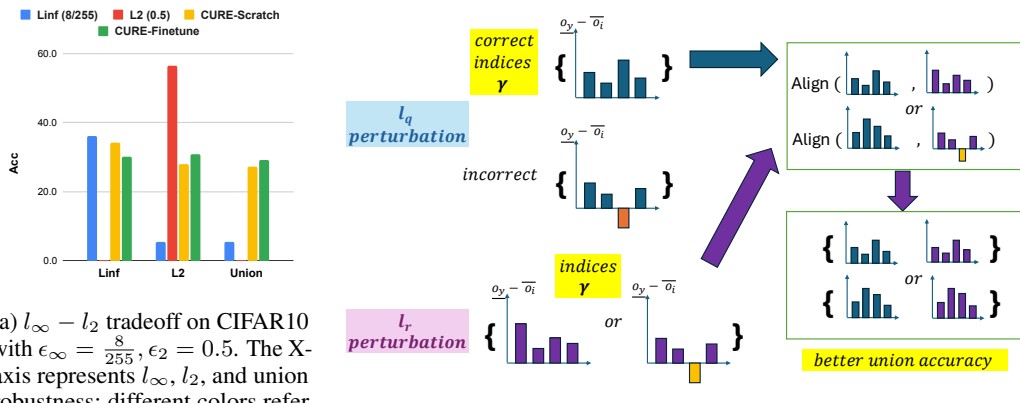

(a) $l_\infty - l_2$ tradeoff on CIFAR10 with $\epsilon_\infty = \frac{8}{255}, \epsilon_2 = 0.5$. The X-axis represents $l_\infty, l_2$, and union robustness; different colors refer to different training methods.

(b) Bound alignment during training.

Figure 1: (a) $l_\infty - l_2$ tradeoff: an $l_\infty$ certified robust model may lack $l_2$ certified robustness and vice versa. **CURE-Scratch** (yellow) and **CURE-Finetune** (green) improve union robustness significantly. (b) We align the output bound differences for $l_q, l_r$ perturbations on the correctly certified $l_q$ subset $\gamma$ to mitigate $l_q - l_r$ tradeoff for better union robustness.

$l_q$ and $l_r$ robustness, where $l_q$ trained model has lower robustness against its own perturbation $l_q$ (e.g. $l_\infty$ certifiably trained model (blue) in Figure 1a). We observe that the $l_q$ robustness is the bottleneck for attaining better union accuracy when training a model from scratch. Thus, as shown in Figure 1b, at a certain epoch during training, we find the subset of input samples $\gamma$ that IBP (Gowal et al., 2018) proves robustly classified with respect to the $l_q$ perturbations. We then propose a new *bound alignment* method to regularize the distributions of output bound differences, computed with IBP, for $l_q, l_r$ perturbations on the correctly certified subset $\gamma$. In this way, we encourage the model *emphasize optimizing* the samples that can potentially become certifiably robust against multi-norm perturbations. Specifically, we use a KL loss to encourage the distributions of the $l_q, l_r$ output bound differences on subset $\gamma$ to be close to each other for better union accuracy. Also, we find that there exist some useful components in natural training that can be extracted and leveraged to improve certified robustness (Jiang & Singh, 2024). To achieve this, we find and incorporate the layer-wise useful natural training components by comparing the similarity of the certified and natural training model updates. Last but not least, we show it is possible to quickly fine-tune an $l_p$ robust model to have superior multi-norm certified robustness using bound alignment. Due to the $l_q - l_r$ tradeoff, bound alignment effectively preserves more $l_q$ robustness when fine-tuning with $l_r$ perturbations, by focusing on the correctly certified subset. Additionally, this technique is useful for quickly attaining multi-norm robustness using wider and diverse model architectures pre-trained with single $l_p$ certified robustness. In Figure 1a, we show that training from scratch (CURE-Scratch) and fine-tuning (CURE-Finetune) significantly improves union robustness compared with single norm training.

**Main Contributions:** Our main contributions are as follows:

- We propose a new $l_2$ deterministic certified training defense without relying on specific model architecture choices, as well as propose multiple methods (CURE-Joint, CURE-Max, CURE-Random) for multi-norm certified training with better union and geometric certified robustness.

- We devise techniques including bound alignment, connecting natural training with certified training, and certified fine-tuning for better union robustness. CURE-Scratch and CURE-Finetune further facilitate the multi-norm certified training procedure and advance multi-norm robustness.

- Compared with SOTA single-norm training method (Müller et al., 2022a), **CURE** improves union robustness up to $22.8\%$ on MNIST, $23.9\%$ on CIFAR-10, and $8.0\%$ on TinyImagenet. Further, it improves robustness against diverse unseen geometric perturbations up to $0.6\%$ on MNIST and $6.8\%$ on CIFAR-10, paving the way to universal certified robustness.

We will publicly release our code upon acceptance of this work.

## 2 RELATED WORK

**Neural network verification.** We consider deterministic verification methods that analyze a neural network by abstract interpretation (Gehr et al., 2018; Singh et al., 2018; 2019a), optimization via linear programming (LP) (De Palma et al., 2021; Wang et al., 2021; Müller et al., 2022b), mixed integer linear programming (MILP) (Tjeng et al., 2017; Singh et al., 2019b), and semidefinite programming (SDP) (Raghunathan et al., 2018; Dathathri et al., 2020). Many of them are *incomplete* methods, sacrificing some precision for better scalability since the neural network verification problem is generally NP-complete (Katz et al., 2017). In our work, we analyze multi-norm certified training using deterministic and incomplete verification methods.

**Certified training.** For $l_\infty$ certified training, a widely-used method IBP (Mirman et al., 2018; Gowal et al., 2018) minimizes a sound over-approximation of the worst-case loss, calculated using the Box relaxation method. Wong et al. (2018) applies DeepZ (Singh et al., 2018) relaxations, estimating using Cauchy random projections. CROWN-IBP (Zhang et al., 2019b) integrates efficient Box propagation with precise linear relaxation-based bounds during the backward pass to estimate the worst-case loss. Balunović & Vechev (2020) consists of a verifier that aims to certify the network using convex relaxation and an adversary that tries to find inputs causing verification to fail. Shi et al. (2021) proposes a new weight initialization method for IBP, adds Batch Normalization (BN) to each layer and designs regularization with a short warmup schedule. Besides this, SABR (Müller et al., 2022a) and TAPS (Mao et al., 2024) are unsound improvements over IBP by connecting IBP to adversarial attacks and adversarial training. For $l_2$ deterministic certified training, recent works (Xu et al., 2022; Hu et al., 2023; 2024) are based on Lipschitz-based certification methods. They design specialized architectures for $l_2$ certified robustness, which is not naturally robust against $l_\infty$ perturbations. In Table 5, we show our $l_2$ certified defense has better $l_2$ robustness compared with Hu et al. (2023) on CIFAR-10. To the best of our knowledge, **CURE** is the first deterministic framework for multi-norm certified robustness, compatible with diverse model architectures.

**Robustness against multiple perturbations.** Adversarial Training (AT) usually employs gradient descent to discover adversarial examples and incorporates them into training for enhanced adversarial robustness (Tramèr et al., 2017; Madry et al., 2017). Numerous works focus on improving robustness (Zhang et al., 2019a; Carmon et al., 2019; Raghunathan et al., 2020; Wang et al., 2020; Wu et al., 2020; Gowal et al., 2020; Zhang et al., 2021; Debenedetti & Troncoso—EPFL, 2022; Peng et al., 2023; Wang et al., 2023) against a *single* perturbation type while remaining vulnerable to other types. Tramer & Boneh (2019); Kang et al. (2019) observe that robustness against $l_p$ attacks does not necessarily transfer to other $l_q$ attacks ($q \neq p$). Previous studies (Tramer & Boneh, 2019; Maini et al., 2020; Madaan et al., 2021; Croce & Hein, 2022; Jiang & Singh, 2024) modified Adversarial Training (AT) to enhance robustness against multiple $l_p$ attacks, employing average-case (Tramer & Boneh, 2019), worst-case (Tramer & Boneh, 2019; Maini et al., 2020; Jiang & Singh, 2024), and random-sampled (Madaan et al., 2021; Croce & Hein, 2022) defenses. There are also works (Nandy et al., 2020; Liu et al., 2020; Xu et al., 2021; Xiao et al., 2022; Maini et al., 2022) that use preprocessing, ensemble methods, mixture of experts, and stability analysis to solve this problem. For multi-norm certified robustness, Nandi et al. (2023) study the certified multi-norm robustness with probabilistic guarantees. They apply randomized smoothing, which is expensive to compute in nature, making it impractical for real-world applications. Our work in contrast to these works, proposes the first *deterministic* certified multi-norm training for better multi-norm and universal certified robustness.

## 3 BACKGROUND

In this section, we provide the necessary background for **CURE**. We consider a standard classification task with samples $\{(x_i, y_i)\}_{i=0}^N$ drawn from a data distribution $\mathcal{D}$. The input consists of images $x \in \mathbb{R}^d$ with corresponding labels $y \in \mathbb{R}^k$. The objective of standard training is to obtain a classifier $f$ parameterized by $\theta$ that minimizes a loss function $\mathcal{L} : \mathbb{R}^k \times \mathbb{R}^k \to \mathbb{R}$ over $\mathcal{D}$.

### 3.1 NEURAL NETWORK VERIFICATION

Neural network verification is used to formally prove the robustness of a neural network. The portion of the samples that can be proved robust is called *certified accuracy*. Box or interval bounded propagation (Gowal et al., 2018; Mirman et al., 2018) (IBP) is a simple yet effective

verification method. Essentially, IBP calculates an over-approximation of the network's reachable set by propagating an over-approximation of the input region $B_p(x, \epsilon_p), p \in \{2, \infty\}$ through the network, and then verifies whether all reachable outputs result in the correct classification. For instance, we consider a network $f = L_j \circ \sigma \circ L_{j-2} \circ \ldots \circ L_1$, with linear layers $L_i$ and ReLU activation functions $\sigma$. We then propagate $B_p(x, \epsilon_p)$ layer by layer (see Gowal et al. (2018); Mirman et al. (2018) for more details). For the output $o = \{\overline{o}_i, \underline{o}_i\}_{i=0}^{i<k}$, the lower bound of the correct class should be higher than the upper bounds of other classes ($\forall i \in [0, k), i \neq y, \overline{o}_i - \underline{o}_y < 0$) to be provably robust.

## 3.2 TRAINING FOR ROBUSTNESS

A classifier is adversarially robust on an $l_p$-norm ball $B_p(x, \epsilon_p) = \{x' \in \mathbb{R}^d : \|x' - x\|_p \leq \epsilon_p\}$ if it classifies all points within the adversarial region as the correct class. That is, $f(x') = y$ for all perturbed inputs $x' \in B_p(x, \epsilon_p)$. Training for robustness is formulated as a min-max optimization problem. Formally, the optimization problem against a specific $l_p$ attack can be expressed as follows:

$$\min_{\theta} \mathbb{E}_{(x,y) \sim \mathcal{D}} \left[ \max_{x' \in B_p(x, \epsilon_p)} \mathcal{L}(f(x'), y) \right] \quad (1)$$

The inner maximization problem is impossible to solve exactly. Thus, it is often under or over-approximated, referred to as adversarial training (Madry et al., 2017; Tramèr et al., 2017) and certified training (Gowal et al., 2018; Müller et al., 2022a), respectively. Further, the optimization described above is specific to certain $p$ values and tends to be vulnerable to other perturbation types. To address this, previous research has introduced various methods to train networks adversarially robust against multiple perturbations ($l_1$, $l_2$, and $l_\infty$) simultaneously. In our work, we concentrate on how to train networks to be *certifiably* robust against multiple $l_p$ ($l_2$, $l_\infty$) perturbations.

## 3.3 CERTIFIED TRAINING

There are two main categories of methods to train certifiably robust models: unsound and sound methods. On the one hand, IBP, a sound method, optimizes the following loss function based on logit differences:

$$\mathcal{L}_{\text{IBP}}(x, y, \epsilon_\infty) = \ln(1 + \sum_{i \neq y} e^{\overline{o}_i - \underline{o}_y})$$

On the other hand, the state-of-the-art certified training methods SABR (Müller et al., 2022a) and TAPs (Mao et al., 2024), sacrifice some soundness to get a more precise approximation, resulting in better standard and certified accuracy. To achieve this, SABR finds an adversarial example $x' \in B_\infty(x, \epsilon_\infty - \tau_\infty)$ and propagates a smaller box region $B_\infty(x', \tau_\infty)$ using the IBP loss, which can be expressed as follows:

$$\mathcal{L}_{l_\infty}(x, y, \epsilon_\infty, \tau_\infty) = \max_{x' \in B_\infty(x, \epsilon_\infty - \tau_\infty)} \mathcal{L}_{\text{IBP}}(x', y, \tau_\infty)$$

Our extensions of multi-norm certified training are based on SABR.

## 3.4 UNION CERTIFIED ACCURACY AND UNIVERSAL CERTIFIED ROBUSTNESS

**Union certified accuracy.** We focus on the union threat model $\Delta = B_2(x, \epsilon_2) \cup B_\infty(x, \epsilon_\infty)$ which requires the DNN to be *certifiably* robust within the $l_2$ and $l_\infty$ adversarial regions simultaneously. Union accuracy is then defined as the robustness against $\Delta_{(i)}$ for each $x_i$ sampled from $\mathcal{D}$. In this paper, similar to the prior works (Croce & Hein, 2022), we use union accuracy as the main metric to evaluate the multi-norm *certified* robustness.

**Universal certified robustness.** We measure the generalization ability of multi-norm certified training to other perturbation types, including rotation, translation, scaling, shearing, contrast, and brightness change of geometric transformations (Balunovic et al., 2019; Yang et al., 2022). We define the average certified robustness across these adversaries as universal certified robustness.

## 4 CURE: MULTI-NORM CERTIFIED TRAINING FOR UNIVERSAL ROBUSTNESS

In this section, we present our multi-norm certified training framework **CURE**. First, we introduce a new deterministic $l_2$ certified training defense. Building on this, we propose several methods for multi-norm certified training against $l_2, l_\infty$ perturbations, which serve as the base instantiations of our framework. After that, we design new techniques to improve union-certified accuracy. We note that the techniques inside **CURE** are applicable to $l_1$ perturbations as well.

### 4.1 CERTIFIED TRAINING FOR MULTIPLE NORMS

**Certified training for $l_2$ robustness.** We propose a new deterministic certified training method against $l_2$ adversarial perturbations, inspired by SABR Müller et al. (2022a). For the specified $\epsilon_2$ and $\tau_2$ values for $l_2$ certified training, we first search for the $l_2$ adversarial examples using standard $l_2$ adversarial attacks (Kim, 2020) $x' \in B_2(x, \epsilon_2 - \tau_2)$ in the slightly truncated $l_2$ ball. After that, we propagate a smaller box region $B_\infty(x', \tau_2)$ using the IBP loss. The loss we optimize can be formulated as follows:

$$\mathcal{L}_{l_2}(x, y, \epsilon_2, \tau_2) = \max_{x' \in B_2(x, \epsilon_2 - \tau_2)} \mathcal{L}_{\text{IBP}}(x', y, \tau_2)$$

**Certified training for multi-norm ($l_2$ and $l_\infty$) robustness.** Based on the work (Tramer & Boneh, 2019; Madaan et al., 2021; Croce & Hein, 2022) on adversarial training for multiple norms, to combine the optimization of $l_2$ and $l_\infty$ certified training, we propose the following methods:

1. **CURE-Joint**: optimizes $\mathcal{L}_{l_\infty}$ and $\mathcal{L}_{l_2}$ together:

$$\mathcal{L}_{Joint} = (1 - \alpha) \cdot \mathcal{L}_{l_\infty}(x, y, \epsilon_\infty, \tau_\infty) + \alpha \cdot \mathcal{L}_{l_2}(x, y, \epsilon_2, \tau_2)$$

From the adversarial example perspective, it takes the sum of two worst-case IBP losses with $l_\infty$ and $l_2$ examples using a convex combination of weights with hyperparameter $\alpha \in [0, 1]$.

2. **CURE-Max**: compares the values of $\mathcal{L}_{l_2}$ and $\mathcal{L}_{l_\infty}$ and takes the one with a worse (higher) IBP loss. It can be viewed as a *worst-case* defense since it considers the worst-case adversarial examples with higher IBP losses generated by the multiple perturbation types. The max loss $\mathcal{L}_{Max}$ is shown as:

$$\mathcal{L}_{Max} = \max_{p \in \{2, \infty\}} \max_{x' \in B_p(x, \epsilon_p - \tau_p)} \mathcal{L}_{\text{IBP}}(x, y, \epsilon_p, \tau_p)$$

3. **CURE-Random**: randomly partitions a batch of data $(\mathbf{x}, \mathbf{y}) \sim \mathcal{D}$ into equal sized blocks $(\mathbf{x}_1, \mathbf{y}_1)$ and $(\mathbf{x}_2, \mathbf{y}_2)$. For $(\mathbf{x}_1, \mathbf{y}_1)$, we calculate the $l_\infty$ worst-case IBP loss $\mathcal{L}_{l_\infty}$ with $l_\infty$ perturbations. For the other half $(\mathbf{x}_2, \mathbf{y}_2)$, similarly, we get the $l_2$ worst-case IBP loss by applying $l_2$ perturbations. After that, we optimize the **Joint** loss of these two with equal weights, as shown below. In this way, we reduce the time cost of propagating the bounds and generating the adversarial examples by $\frac{1}{2}$.

$$\mathcal{L}_{Random} = \mathcal{L}_{l_\infty}(\mathbf{x}_1, \mathbf{y}_2, \epsilon_\infty, \tau_\infty) + \mathcal{L}_{l_2}(\mathbf{x}_2, \mathbf{y}_2, \epsilon_2, \tau_2), \text{where } \mathbf{x} = \mathbf{x}_1 \cup \mathbf{x}_2, \mathbf{y} = \mathbf{y}_1 \cup \mathbf{y}_2$$

### 4.2 IMPROVED MULTI-NORM CERTIFIED TRAINING

The methods proposed above are suboptimal as they fail to fully explore the relationship between worst-case IBP losses across different perturbations, certified training (CT), and natural training (NT). To address this, we introduce the following improvements to enhance the union robustness of **CURE**: (1) We identify a tradeoff between $l_2$ and $l_\infty$ perturbations and propose a bound alignment technique to mitigate this, improving multi-norm robustness. (2) We analyze and connect certified and natural training to attain better union accuracy. (3) To facilitate the procedure for multi-norm robustness with pre-trained single norm models, we propose the first certified fine-tuning method, demonstrating its ability to quickly improve union accuracy using **CURE** (Table 1).

**Bound alignment (BA) for better union robustness.** As shown in Figure 1a and Table 1, an $l_\infty$ certifiably robust model usually has low $l_2$ certified robustness and vice versa, which reveals that there

exists a tradeoff between $l_2$ and $l_\infty$ certified robustness. To this end, we investigate the $l_q - l_r$ tradeoff, which provides important intuitions for the design of bound alignment. For given values of $\epsilon_\infty$ and $\epsilon_2$, we aim to achieve good union robustness when performing multi-norm training from scratch on a model $f$. We denote $A_u$ as the optimal union accuracy we can get with multi-norm training. Further, we define and compare the best possible $l_q, l_r$ robustness ($A_q$ and $A_r$ with Definition 4.1) with $l_q$ and $l_r$ certified defenses. In practice, we use our $l_\infty$ and $l_2$ certified defenses to approximate $A_q$ and $A_r$.

**Definition 4.1** ($l_p$ robustness $A_p$). *Given an $\epsilon_p \in \mathbb{R}$ value, we define $l_p$ robustness, denoted as $A_p$, which is the final certified robustness against $l_p$ perturbations for a model that is fully trained using the best $l_p$ certified training strategy.*

Without the loss of generality, we assume $A_q \leq A_r$. In other words, we select $q$ and $r$ values $\in \{\infty, 2\}$ based on the empirical values we get for $A_q$ and $A_r$ using our $l_\infty$ and $l_2$ certified defenses. $A_u$ is upper bounded by $A_q$ since in the most ideal case, the model is robust against $l_q$ and $l_r$ perturbations on the same set of images with union accuracy $A_u = A_q$. Thus, to obtain better union accuracy, the goal is to have $A_u \to A_q$. How do we achieve this? Generally speaking, given a randomly initialized model $f$, we want it to *focus on optimizing* the samples which can potentially be *robust towards both* when training from scratch. Here, we take a closer look at a single training step of $f$'s optimization. During this step, we find the correctly certified $l_q$ subset $\gamma$ of $f$ (Definition 4.2), meaning the subset $\gamma$ for which the lower bound computed with IBP of the correct class is higher than the upper bounds of other classes.

**Definition 4.2** (Correctly Certified $l_q$ Subset). *At epoch $e$, given the perturbation size $\epsilon_q \in \mathbb{R}$ and model $f$, for a batch of data $(\mathbf{x}, \mathbf{y}) \sim \mathcal{D}$ with size $n$, we have the output upper and lower bounds computed by IBP for $l_q$ perturbations. We define a function $h$ for this procedure as $h(\mathbf{x}) = \{\overline{\mathbf{o}}_j, \underline{\mathbf{o}}_j\}_{j=0}^{j<n}$, where $\mathbf{o} = \{o_i\}_{i=0}^{i<k}$ is a vector of bounds for all classes. Then, the correctly certified subset $\gamma$ at the current step is defined as:*

$$\forall j \in \gamma \text{ with } (\mathbf{x}_j, \mathbf{y}_j) \text{ and bounds } \{\overline{\mathbf{o}}_j = \{\overline{o}_i\}_{i=0}^{i<k}, \underline{\mathbf{o}}_j = \{\underline{o}_i\}_{i=0}^{i<k}\}, \text{ we have } \forall i \neq y_j, \overline{o}_i \leq \underline{o}_{y_j}.$$

Since $A_q$ serves as the upper bound of $A_u$, similarly, $\gamma$ can be regarded as the subset of inputs that are *more likely* to be optimized for both $l_q$ and $l_r$ robustness. For certified training, people usually optimize the model using bound differences $\{\overline{o}_i - \underline{o}_y\}_{i=0}^{i<k}$ ($y$ is the correct class). Therefore, for better union robustness $A_u$, we align the bound differences $\{\{\overline{o}_i - \underline{o}_y\}_{i=0, i\neq y}^{i<k}\}_{j=0}^{j<n}$ of $l_r$ and $l_q$ certified training outputs, specifically on the correctly certified $l_q$ subset $\gamma$. Specifically, for each batch of data $(\mathbf{x}, \mathbf{y}) \sim \mathcal{D}$, we generate predicted bounds $h(\mathbf{x}, q) = \{\overline{\mathbf{o}}_{qj}, \underline{\mathbf{o}}_{qj}\}_{j=0}^{j<n}$ and $h(\mathbf{x}, r) = \{\overline{\mathbf{o}}_{rj}, \underline{\mathbf{o}}_{rj}\}_{j=0}^{j<n}$. We denote their bounds differences after softmax normalization as $d_q = \{\{\overline{o}_{qi} - \underline{o}_{qy}\}_{i=0, i\neq y}^{i<k}\}_{j=0}^{j<n}$ and $d_r = \{\{\overline{o}_{ri} - \underline{o}_{ry}\}_{i=0, i\neq y}^{i<k}\}_{j=0}^{j<n}$. Then, we select indices $\gamma$, according to Definition 4.2. We denote the size of the indices as $n_c$. We compute a KL-divergence loss over this set of samples using $KL(d_q[\gamma]\|d_r[\gamma])$ (Eq. 2). Intuitively, we want to make $d_r[\gamma]$ and $d_q[\gamma]$ distributions close to each other, such that we gain more union robustness by regularizing the model to optimize more on the subset of examples which potentially brings $A_u \to A_q$.

$$\mathcal{L}_{KL} = \frac{1}{n_c} \cdot \sum_{i=1}^{n_c} \sum_{j=0}^{k} d_q[\gamma[i]][j] \cdot \log\left(\frac{d_q[\gamma[i]][j]}{d_r[\gamma[i]][j]}\right) \tag{2}$$

Apart from the KL loss, we add another loss term using a Max-style approach in Eq. 3, since Max has relatively good performance and small computational cost, as shown in Table 1 and Table 6. $\mathcal{L}_{Max}$ is the worst-case loss between $l_r, l_q$ certified training with IBP. Our final loss $\mathcal{L}_{\text{Scratch}}$ combines $\mathcal{L}_{KL}$ and $\mathcal{L}_{Max}$, via a hyper-parameter $\eta$ in Eq. 4.

$$\mathcal{L}_{Max} = \max_{p \in \{2, \infty\}} \max_{x' \in B_p(x, \epsilon_p - \tau_p)} \mathcal{L}_{\text{IBP}}(x, y, \epsilon_p, \tau_p) \quad (3) \qquad \mathcal{L}_{\text{Scratch}} = \mathcal{L}_{Max} + \eta \cdot \mathcal{L}_{KL} \quad (4)$$

**Integrate natural training (NT) into certified training (CT).** In the context of adversarial robustness, Jiang & Singh (2024) shows that there exist some useful components in natural training, which can be extracted and properly integrated into adversarial training to get better adversarial robustness. We propose a technique to integrate NT into certified training (CT), to enhance union-certified robustness.

To effectively connect NT with CT, we analyze the training procedures of the two. Specifically, for model $f^{(r)}$ at any epoch $r$, we examine the model updates of NT and CT over all samples from $\mathcal{D}$. The models $f_n^{(r)}$ and $f_c^{(r)}$ represent the results after one epoch of natural training and certified training using $\mathcal{L}_{\text{Scratch}}$, respectively, both beginning from the same initial model $f^{(r)}$. Then we compare the natural updates $g_n = f_n^{(r)} - f^{(r)}$ and certified updates $g_c = f_c^{(r)} - f^{(r)}$. Our goal is to identify useful components from $g_n$ and incorporate them into $g_c$ for better certified robustness. For a specific layer $l$, comparing $g_n^l$ and $g_c^l$, we retain a portion of $g_n^l$ according to their cosine similarity score (Eq.5). Negative scores indicate that $g_n^l$ does not contribute to certified robustness, so we discard components with similarity scores $\leq 0$. The **GP** (Gradient Projection) operation, defined in Eq.6, projects $g_c^l$ towards $g_n^l$.

$$\cos(g_n^l, g_c^l) = \frac{g_n^l \cdot g_c^l}{\|g_n^l\|\|g_c^l\|} \qquad (5) \qquad \mathbf{GP}(g_n^l, g_c^l) = \begin{cases} \cos(g_n^l, g_c^l) \cdot g_n^l, & \cos(g_n^l, g_c^l) > 0 \\ 0, & \cos(g_n^l, g_c^l) \leq 0 \end{cases} \qquad (6)$$

Therefore, the total projected (useful) model updates $g_p$ coming from $g_n$ could be computed as Eq. 7. We use $\mathcal{M}$ to represent all layers of the current model update. The expression $\bigcup_{l \in \mathcal{M}}$ concatenates the useful natural model update components from all layers. A hyper-parameter $\beta$ is introduced to balance the contributions of $g_{GP}$ and $g_c$, as outlined in Eq.8. It is important to note that this projection procedure is applied only after certified training with the full epsilon value.

$$g_p = \bigcup_{l \in \mathcal{M}} \mathbf{GP}(g_n^l, g_c^l) \qquad (7) \qquad f^{(r+1)} = f^{(r)} + \beta \cdot g_p + (1 - \beta) \cdot g_c \qquad (8)$$

**Quick certified fine-tuning of single-norm pre-trained classifiers for multi-norm robustness.** In practice, as the model architectures and datasets become larger, multi-norm certified training from scratch becomes more expensive. Also, there are many pre-trained models available with single norm certified training. In adversarial robustness, Croce & Hein (2022) shows it is possible to obtain state-of-the-art multi-norm robustness by fine-tuning a pre-trained model for a few epochs, which reduces the computational cost significantly. In this work, we propose the first fine-tuning certified multi-norm robustness scheme **CURE-Finetune**. Starting from a single norm pre-trained model, we perform the bound alignment technique by optimizing $\mathcal{L}_{\text{Scratch}}$ for a few epochs. Because of the $l_q - l_r$ tradeoff, certifiably finetuning a $l_q$ pre-trained model on $l_r$ perturbations reduces $l_q$ robustness. Thus, we want to preserve more $l_q$ robustness when doing certified fine-tuning, which makes bound alignment useful here. By regularizing on the correctly certified $l_q$ subset with $\mathcal{L}_{\text{Scratch}}$, we can prevent losing more $l_q$ robustness when boosting $l_r$ robustness, which leads to better union accuracy. We note that **CURE-Finetune** can be adapted to any single-norm certifiably pre-trained models. As shown in Table 1, compared with other methods, we can quickly obtain a superior multi-norm certified robustness by performing fine-tuning on pre-trained $l_\infty$ models for a few epochs.

## 5 EXPERIMENT

In this section, we present and discuss the results of union robustness, geometric robustness, and ablation studies on hyper-parameters for MNIST, CIFAR-10, and TinyImagenet experiments. Other ablation studies, visualizations, and algorithms of **CURE** can be found in Appendix B and D.

**Experimental Setup.** For datasets, we use MNIST (LeCun et al., 2010) and CIFAR-10 (Krizhevsky et al., 2009) which both include 60K images with 50K and 10K images for training and testing, as well as TinyImageNet (Le & Yang, 2015) which consists of 200 object classes with 500 training images, 50 validation images, and 50 test images per class. We compare the following methods: 1. $l_\infty$: SOTA $l_\infty$ certified defense SABR (Müller et al., 2022a), 2. $l_2$: our proposed $l_2$ certified defense, 3. **CURE-Joint**: take a weighted sum of $l_2, l_\infty$ IBP losses. 4. **CURE-Max**: take the worst of $l_2, l_\infty$ IBP losses. 5. **CURE-Random**: randomly partitions the samples into two blocks, then applies the Joint loss with equal weights. 6. **CURE-Scratch**: training from scratch with bound alignment and gradient projection techniques. 7. **CURE-Finetune**: robust fine-tuning with the bound alignment technique using $l_\infty$ pre-trained models. We use a 7-layer convolutional architecture CNN7 similar to prior work (Müller et al., 2022a) for models. In Table 5, we compare our proposed $l_2$ defense with Hu et al. (2023), where we show our method outperforms the SOTA $l_2$ deterministic certified defense on CIFAR-10. We choose similar hyperparameters and training setup as Müller et al. (2022a)

for $l_\infty$ certified training. We select $\alpha = 0.5$, $l_2$ subselection ratio $\lambda_2 = 1e^{-5}$, $\beta = 0.8$, and $\eta = 2.0$ according to our ablation study results in Section 5.2 and Appendix B. For robust fine-tuning, we finetune $20\%$ of the original epochs from scratch. More implementation details are in Appendix A.

**Evaluation.** We choose the common $\epsilon_\infty, \epsilon_2$ values used in the literature (Müller et al., 2022a; Hu et al., 2023) to construct multi-norm regions. These include $(\epsilon_2 = 0.5, \epsilon_\infty = 0.1), (\epsilon_2 = 1.0, \epsilon_\infty = 0.3)$ for MNIST, $(\epsilon_2 = 0.25, \epsilon_\infty = \frac{2}{255}), (\epsilon_2 = 0.5, \epsilon_\infty = \frac{8}{255})$ for CIFAR-10 and $(\epsilon_2 = \frac{36}{255}, \epsilon_\infty = \frac{1}{255})$ for TinyImageNet. We make sure the adversarial regions with sizes $\epsilon_\infty$ and $\epsilon_2$ do not include each other. We report the clean accuracy, certified accuracy against $l_2, l_\infty$ perturbations, union accuracy, and individual/average certified robustness against geometric transformations. Further, we use alpha-beta crown (Zhang et al., 2018) for certification on $l_2, l_\infty$ perturbations and FGV (Yang et al., 2022) for efficient certification of geometric transformations.

## 5.1 Main Results

| Dataset | $(\epsilon_\infty, \epsilon_2)$ | Methods | Clean | $l_\infty$ | $l_2$ | Union |
|---|---|---|---|---|---|---|
| MNIST | (0.1, 0.5) | $l_\infty$ | 99.2 | 97.7 | 96.9 | 96.9 |
| | | $l_2$ | 99.5 | 2.0 | 98.7 | 2.0 |
| | | CURE-Joint | 99.3 | 97.5 | 97.4 | 97.1 |
| | | CURE-Max | 99.3 | 97.5 | 97.5 | **97.4** |
| | | CURE-Random | 99.2 | 96.9 | 96.7 | 96.6 |
| | | CURE-Scratch | 99.0 | 97.3 | 97.5 | **97.2** |
| | | CURE-Finetune | 99.1 | 96.9 | 97.5 | 96.9 |
| | (0.3, 1.0) | $l_\infty$ | 99.0 | 91.0 | 64.5 | 62.9 |
| | | $l_2$ | 99.4 | 0.0 | 63.0 | 0.0 |
| | | CURE-Joint | 98.7 | 89.8 | 78.3 | 75.7 |
| | | CURE-Max | 98.7 | 91.1 | 76.2 | 74.8 |
| | | CURE-Random | 98.6 | 90.2 | 78.9 | 77.0 |
| | | CURE-Scratch | 98.0 | 89.4 | 85.9 | 83.9 |
| | | CURE-Finetune | 98.6 | 90.0 | 90.0 | **85.7** |
| CIFAR-10 | $(\frac{2}{255}, 0.25)$ | $l_\infty$ | 79.4 | 59.7 | 67.8 | 59.7 |
| | | $l_2$ | 82.3 | 5.6 | 71.2 | 5.6 |
| | | CURE-Joint | 80.2 | 57.3 | 69.7 | 57.3 |
| | | CURE-Max | 77.7 | 59.6 | 68.2 | 59.6 |
| | | CURE-Random | 78.9 | 57.5 | 68.3 | 57.5 |
| | | CURE-Scratch | 76.9 | 60.9 | 67.8 | **60.9** |
| | | CURE-Finetune | 78.0 | 59.7 | 68.2 | 59.7 |
| | $(\frac{8}{255}, 0.5)$ | $l_\infty$ | 51.0 | 36.1 | 5.4 | 5.4 |
| | | $l_2$ | 73.3 | 0.0 | 56.6 | 0.0 |
| | | CURE-Joint | 52.1 | 22.1 | 30.5 | 20.0 |
| | | CURE-Max | 51.5 | 33.9 | 19.5 | 18.8 |
| | | CURE-Random | 52.4 | 28.4 | 30.5 | 24.3 |
| | | CURE-Scratch | 49.5 | 34.2 | 28.1 | 27.3 |
| | | CURE-Finetune | 40.2 | 30.2 | 30.8 | **29.3** |
| TinyImagnet | $(\frac{1}{255}, \frac{36}{255})$ | $l_\infty$ | 28.3 | 20.1 | 24.6 | 16.1 |
| | | $l_2$ | 36.1 | 2.5 | 29.7 | 2.5 |
| | | CURE-Joint | 29.4 | 22.6 | 24.6 | 22.6 |
| | | CURE-Max | 28.8 | 22.1 | 24.6 | 22.1 |
| | | CURE-Random | 30.1 | 22.1 | 24.6 | 22.1 |
| | | CURE-Scratch | 28.1 | 24.1 | 25.1 | **24.1** |
| | | CURE-Fintune | 27.9 | 19.1 | 23.1 | 19.1 |

Table 1: Comparison of the clean accuracy, as well as individual, and union certified accuracy (%) for different multi-norm certified training methods. **CURE** consistently improves union accuracy compared with single-norm training with significant margins on all datasets. **CURE-Scratch** and **CURE-Finetune** outperform other methods in most cases.

**Union accuracy on MNIST, CIFAR-10, and TinyImagenet with CURE framework.** In Table 1, we show the results of clean accuracy and certified robustness against single and multi-norm with **CURE** on MNIST, CIFAR-10, and TinyImagenet. We observe that CURE-Joint, CURE-Max, and CURE-Random usually result in better union robustness compared with $l_2$ and $l_\infty$ certified

training. Further, **CURE-Scratch** and **CURE-Finetune** consistently improve the union accuracy compared with other multi-norm methods with significant margins in most cases ($10\%$ for MNIST ($\epsilon_\infty = 0.3, \epsilon_2 = 1.0$), $3 - 10\%$ for CIFAR-10 ($\epsilon_\infty = \frac{8}{255}, \epsilon_2 = 0.5$), and $2\%$ for TinyImagenet experiments), showing the effectiveness of bound alignment and gradient projection techniques. Also, for quick fine-tuning, we show it is possible to quickly fine-tune a $l_\infty$ robust model with good union robustness using bound alignment, achieving SOTA union accuracy on MNIST ($\epsilon_\infty = 0.3, \epsilon_2 = 1.0$) and CIFAR-10 ($\epsilon_\infty = \frac{8}{255}, \epsilon_2 = 0.5$) experiments.

**Robustness against geometric transformations.** Table 2 and Figure 2 compare **CURE** with single norm training against various geometric perturbations on MNIST and CIFAR-10 datasets. For both experiments, **CURE** outperforms single norm training on diverse geometric transformations ($0.6\%$ for MNIST and $6\%$ for CIFAR-10 on average), leading to better *universal certified robustness*. Also, **CURE-Scratch** has better geometric robustness than **CURE-Max** on both datasets, which reveals that bound alignment and gradient projection lead to better universal certified robustness.

| Configs | R(30) | $T_u(2),T_v(2)$ | Sc(5),R(5), C(5),B(0.01) | Sh(2),R(2),Sc(2), C(2),B(0.001) | Avg |
|---|---|---|---|---|---|
| $l_\infty$ | 54.6 | 20.9 | 82.5 | 95.6 | 63.4 |
| $l_2$ | 0.0 | 0.0 | 0.0 | 0.0 | 0.0 |
| CURE-Joint | **55.9** | 21.3 | 82.3 | **95.7** | 63.8 |
| CURE-Max | 50.1 | 20.7 | 80.2 | 94.8 | 61.5 |
| CURE-Random | 54.8 | 18.8 | 83.5 | 95.6 | 63.2 |
| CURE-Scratch | 51.0 | **24.3** | **85.5** | 95.1 | **64.0** |

Table 2: Comparison on **CURE** against geometric transformations for MNIST experiment. We denote $R(\varphi)$ a rotation of $\pm\varphi$ degrees; $T_u(\Delta u)$ and $T_v(\Delta v)$ a translation of $\pm\Delta u$ pixels horizontally and $\pm\Delta v$ pixels vertically, respectively; $Sc(\lambda)$ a scaling of $\pm\lambda\%$; $Sh(\gamma)$ a shearing of $\pm\gamma\%$; $C(\alpha)$ a contrast change of $\pm\alpha\%$; and $B(\beta)$ a brightness change of $\pm\beta$. **CURE** improves the average robustness compared with single norm training with better geometric certified robustness. Also, **CURE-Scratch** achieves the best average geometric transformation robustness.

| Configs | R(10) | R(2),Sh(2) | Sc(1),R(1), C(1),B(0.001) | Avg |
|---|---|---|---|---|
| $l_\infty$ | 27.8 | 33.2 | 23.3 | 28.1 |
| $l_2$ | 36.6 | 0.0 | 0.0 | 12.2 |
| Joint | **35.0** | **41.4** | **28.2** | **34.9** |
| Max | 33.7 | 39.0 | 23.3 | 32.0 |
| Random | 35.1 | 40.9 | 26.2 | 34.1 |
| Scratch | 34.2 | 39.6 | 24.9 | 32.9 |

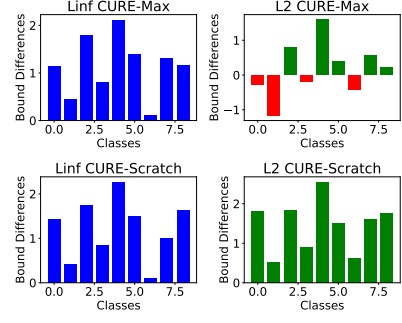

Figure 2: Comparison on **CURE** against geometric transformations for CIFAR-10 experiment. **CURE** improves the universal certified robustness significantly compared with single norm training.

Figure 3: CURE-Max and CURE-Scratch bound difference visualization.

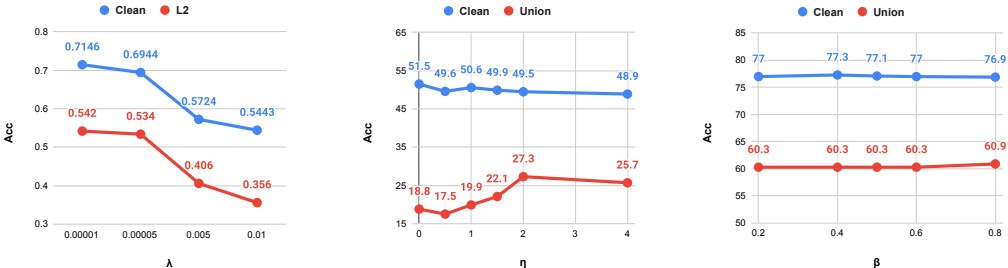

(a) $\lambda_2$: subselection ratio for $l_2$.  (b) $\eta$: weight for bound alignment.  (c) $\beta$: hyper-parameter for GP.

Figure 4: Alabtion studies on $\lambda_2$, $\eta$ and $\beta$ hyper-parameters.

## 5.2 ABLATION STUDY

**Subselection ratio** $\lambda$. For $l_\infty$ certified training, we use the same $\lambda_\infty$ as in Müller et al. (2022a). For $l_2$ subselection ratio $\lambda_2$, in Figure 4a, we show the $l_2$ certified robustness using varying $\lambda_2 \in [1e^{-5}, 5e^{-5}, 5e^{-3}, 1e^{-2}]$ with $\epsilon_2 = 0.5$. Both clean and $l_2$ accuracy improves when we have smaller $\tau_2$ values. Based on the results, we choose $\tau_2 = 1e^{-5}$ for our experiments.

**Bound alignment (BA) hyper-parameter** $\eta$. We perform CIFAR-10 ($\epsilon_\infty = \frac{8}{255}, \epsilon_2 = 0.5$) experiments with different $\eta$ values in $[0.5, 1.0, 1.5, 2.0, 4.0]$. In Figure 4b, we observe that the clean accuracy generally drops as we have larger $\eta$ values, with union accuracy improving then dropping. We pick $\eta = 2.0$ with the best union accuracy for the experiments.

**Gradient projection (GP) hyper-parameter** $\beta$. Figure 4c displays the sensitivity of clean and union accuracy with different choices of $\beta$ values on CIFAR-10 ($\epsilon_\infty = \frac{2}{255}, \epsilon_2 = 0.25$) experiments. CURE-Scratch is generally insensitive to varying $\beta$ values. We choose $\beta = 0.8$ for the experiments to be relatively the best.

**Ablation study on BA and GP.** In Table 3, we show the ablation study of BA and GP techniques on the MNIST ($\epsilon_\infty = 0.3, \epsilon_2 = 1.0$) experiment. BA and GP improve union accuracy by 2% and 7% respectively, demonstrating the individual effectiveness of our proposed techniques.

**Visualization of bound differences.** Figure 3 displays the bound differences $\{o_y - \bar{o}_i\}_{i=0, i\neq y}^{i<k}$ of one example that is improved by **CURE-Scratch** (second row), compared with the **CURE-Max** (first row), from the CIFAR-10 ($\epsilon_\infty = \frac{8}{255}, \epsilon_2 = 0.5$) experiment with $q = \infty, r = 2$. We use the outputs from

|          | Clean | $l_\infty$ | $l_2$ | Union |
|----------|-------|-----------|-------|-------|
| CURE-Max | 98.7  | 91.1      | 76.2  | 74.8  |
| +BA      | 98.6  | 91.0      | 78.2  | 76.5  |
| +BA + GP | 98.0  | 89.4      | 85.9  | **83.9** |

Table 3: Ablations on BA and GP.

alpha-beta-crown. For $l_2$ perturbations (blue diagrams), we demonstrate that **CURE-Scratch** exhibits all positive bound differences, whereas **CURE-Max** shows several negative bound differences (highlighted in red), leading to a robust union prediction. Additionally, we observe that the distributions in the second row are more aligned than those in the first row. **CURE-Scratch** effectively aligns the bound difference distributions when the model is robust against $l_q$ perturbations, bringing the distributions closer together. This demonstrates the effectiveness of the bound alignment method. Additional visualizations are available in Appendix B.

## 5.3 DISCUSSIONS

**Time cost of CURE.** The extra training costs of GP are small, taking $6, 24, 82$ seconds using a single NVIDIA A40 GPU on MNIST, CIFAR-10, and TinyImageNet datasets (Table 7), respectively. Compared with the total training cost of **CURE-Scratch**, it only accounts for $\sim 6\%$ of the total cost. For runtime comparison of different methods, we have a complete runtime analysis (Table 6) in Appendix C for the MNIST experiment. We observe that **CURE-Joint** has around two times the cost of other methods. **CURE-Finetune** has the smallest time cost per epoch, which shows the efficiency of our proposed techniques.

**Limitations.** For $l_2$ certified training, we use a $l_\infty$ box instead of $l_2$ ball for bound propagation, which leads to more over-approximation and the potential loss of precision. Also, we notice drops in clean accuracy in both training from scratch and fine-tuning with **CURE** methods. In some cases, union accuracy improves slightly but clean accuracy and single $l_p$ robustness reduce. Both BA and GP techniques lead to a slight decrease in clean accuracy on experiments of three datasets. There is no negative societal impact of this work.

## 6 CONCLUSION

We propose the first framework **CURE** with a new $l_2$ deterministic certified defense and multi-norm training methods for better union robustness. Further, we devise bound alignment, gradient projection, and robust certified fine-tuning techniques, to enhance and facilitate the union-certified robustness. Extensive experiments on MNIST, CIFAR-10, and TinyImagenet show that **CURE** significantly improves both union accuracy and robustness against geometric transformations, paving the path to universal certified robustness.

## 7 REPRODUCIBILITY STATEMENT

We provide the source code of **CURE** as part of the supplementary material that can be used to reproduce our results. We provide the details of our hyper-parameters, training scheme, and model architecture in Section 5. We also provide additional details including other training details, further evaluation, and pseudocode not covered in the main text in the appendix.

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

## A    MORE TRAINING DETAILS

We mostly follow the hyper-parameter choices from Müller et al. (2022a) for **CURE**. We include weight initialization and warm-up regularization from Shi et al. (2021). Further, we use ADAM (Kingma, 2014) with an initial learning rate of $1e^{-4}$, decayed twice with a factor of 0.2. For CIFAR-10, we train 160 and 180 epochs for ($\epsilon_\infty = \frac{2}{255}, \epsilon_2 = 0.25$) and ($\epsilon_\infty = \frac{8}{255}, \epsilon_2 = 0.5$), respectively. We decay the learning rate after 120 and 140, 140 and 160 epochs, respectively. For the TinyImagenet experiment, we use the same setting as ($\epsilon_\infty = \frac{8}{255}, \epsilon_2 = 0.5$). For the MNIST dataset, we train 70 epochs, decaying the learning rate after 50 and 60 epochs. For batch size, we set 128 for CIFAR-10 and TinyImagenet and 256 for MNIST. For all experiments, we first perform one epoch of standard training. Also, we anneal $\epsilon_\infty, \epsilon_2$ from 0 to their final values with 80 epochs for CIFAR-10 and TinyImagenet and 20 epochs for MNIST. We only apply GP after training with the final epsilon values. For certification, we verify 1000 examples on MNIST and CIFAR-10, as well as 199 examples on TinyImagenet.

## B    OTHER ABLATION STUDIES

**Hyper-parameter $\alpha$ for Joint certified training.** As shown in Table 4, we test the changing of $l_\infty, l_2$, and union accuracy with different $\alpha$ values in $[0, 0.25, 0.5, 0.75, 1.0]$ on MNIST ($\epsilon_\infty = 0.1, \epsilon_2 = 0.5$) experiments. We observe that $\alpha = 0.5$ has the best union accuracy and is generally a good choice for our experiments by balancing the two losses.

| $\alpha$ | 0.0 | 0.25 | 0.5 | 0.75 | 1.0 |
|---|---|---|---|---|---|
| Clean | 99.2 | 99.2 | 99.3 | 99.2 | 99.5 |
| $l_\infty$ | 97.7 | 97.7 | 97.5 | 97.2 | 2.0 |
| $l_2$ | 96.9 | 95.6 | 97.4 | 95.9 | 98.7 |
| Union | 96.9 | 95.6 | **97.1** | 95.8 | 2.0 |

Table 4: Ablation study on Joint training hyper-parameter $\alpha$.

**Comparison of $l_2$ certified robustness on $l_2$ deterministic certified training methods.** In Table 5, we compare our proposed $l_2$ certified defense with SOTA $l_2$ certified defense Hu et al. (2023) on CIFAR-10 with $\epsilon_2 = 0.25$ and 0.5. The results show that our proposed $l_2$ deterministic certified training method improves over $l_2$ robustness by $2 \sim 4\%$ compared with the SOTA method.

| $\epsilon_2$ | 0.25 | 0.5 |
|---|---|---|
| Hu et al. (2023) | 69.5 | 52.2 |
| Ours | **71.2** | **56.6** |

Table 5: Comparison of $l_2$ certified accuracy: our proposed $l_2$ certified training consistently outperforms Hu et al. (2023) by $2 \sim 4\%$.

**More visualizations on bound differences.** We plot the bound difference examples from alpha-beta-crown on MNIST, CIFAR-10, and TinyImagenet datasets, where the negative bound differences are colored in red. As shown in Figure 5, 6, 7, 8, 9, we compare CURE-Scratch (second row) with CURE-Max (first row), with bound differences against $l_\infty$ and $l_2$ perturbations colored in blue and green, respectively. CURE-Scratch produces all positive bound differences, leading to unionly robust predictions; CURE-Max is not unionly robust due to some negative bound differences. Also, we observe that CURE-Scratch successfully brings $l_q, l_r$ bound difference distributions close to each other compared with CURE-Max in many cases, which confirms the effectiveness of our bound alignment technique.

## C    RUNTIME ANALYSIS

This section provides the runtime per training epoch for all methods on MNIST ($\epsilon_\infty = 0.1, \epsilon_2 = 0.75$) experiments and runtime per training epoch of CURE-Scratch with ablation studies on GP for MNIST,

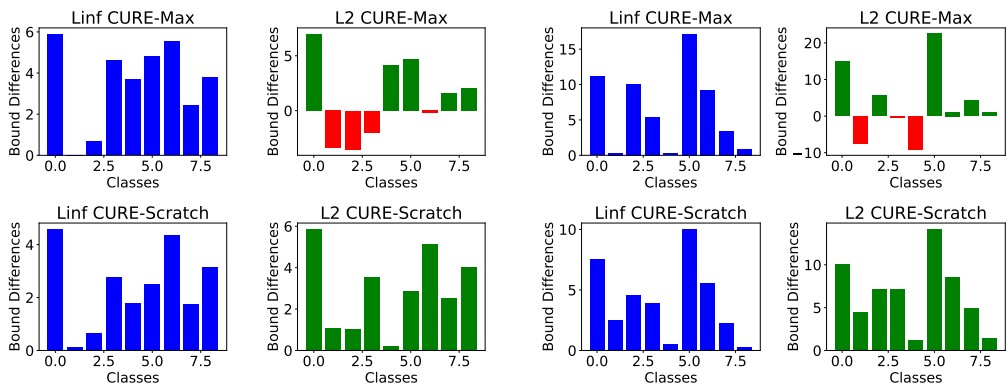

Figure 5: Bound difference visualizations on MNIST ($\epsilon_\infty = 0.3, \epsilon_2 = 1.0$) experiments.

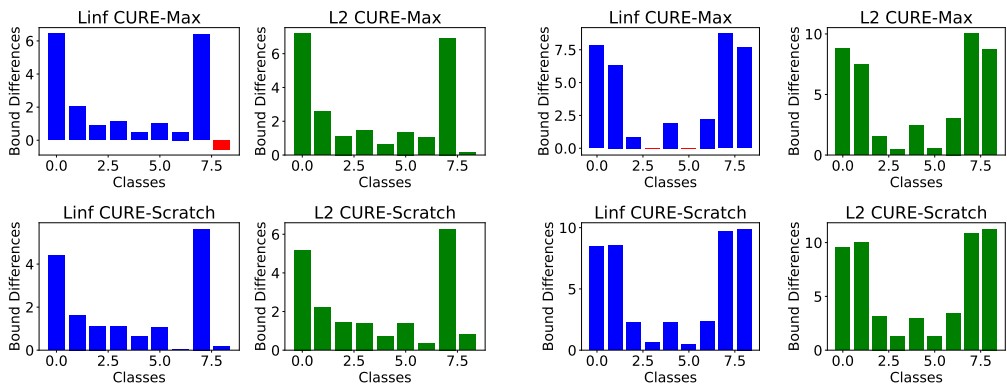

Figure 6: Bound difference visualizations on CIFAR-10 ($\epsilon_\infty = \frac{2}{255}, \epsilon_2 = 0.25$) experiments.

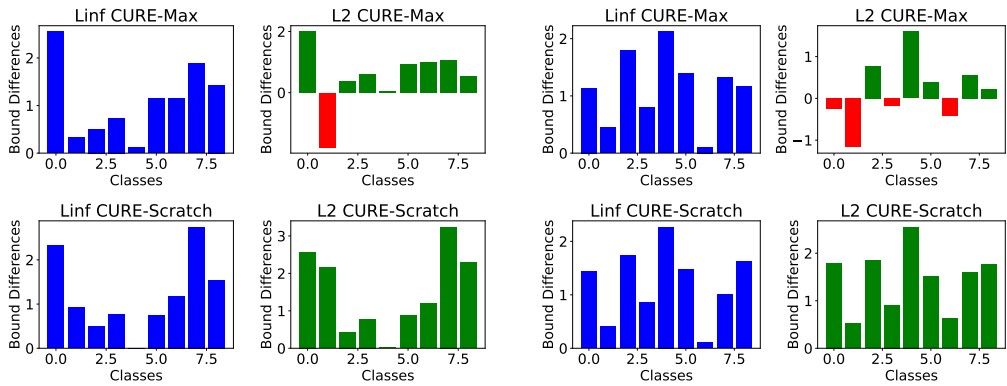

Figure 7: Bound difference visualizations on CIFAR-10 ($\epsilon_\infty = \frac{8}{255}, \epsilon_2 = 0.5$) experiments.

CIFAR10, and TinyImagenet experiments. We evaluate all the methods on a single A40 Nvidia GPU with 40GB memory and the runtime is reported in seconds (s).

**Runtime for different methods on MNIST experiments.** In Table 6, we show the time in seconds (s) per training epoch for single norm training ($l_\infty$ and $l_2$), CURE-Joint, CURE-Max, CURE-Random, CURE-Scratch, and CURE-Finetune methods. CURE-Finetune has the smallest training cost compared with other methods and CURE-Joint has the highest time cost (around two times of other methods) per epoch. The results indicate the efficiency of training with CURE-Scratch/Finetune.

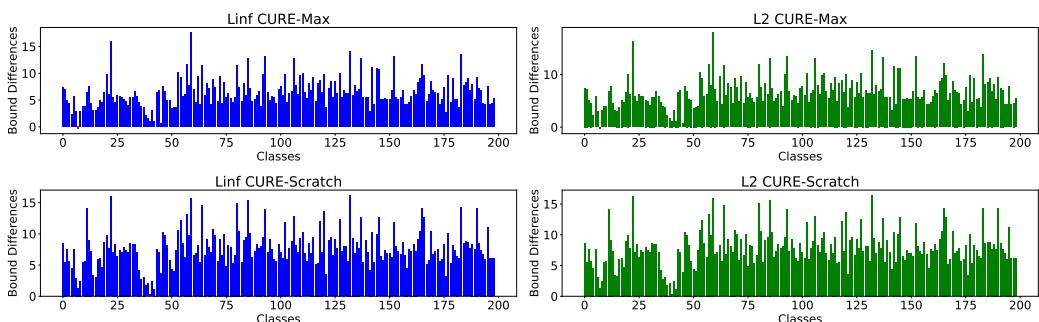

Figure 8: Bound difference visualization on TinyImagenet ($\epsilon_\infty = \frac{1}{255}, \epsilon_2 = \frac{36}{255}$) experiments.

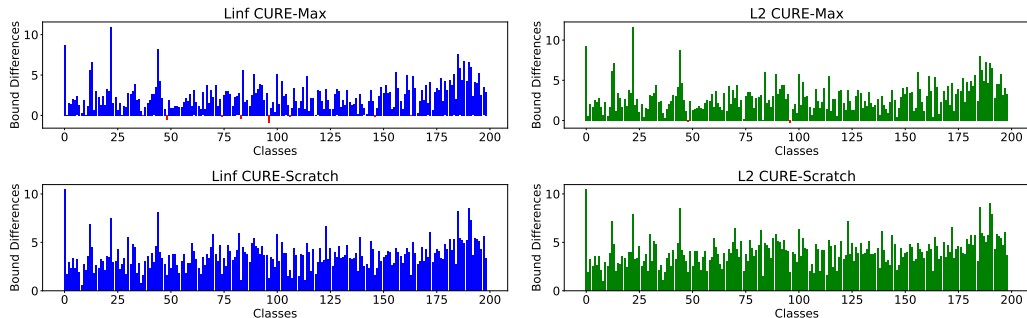

Figure 9: Bound difference visualization on TinyImagenet ($\epsilon_\infty = \frac{1}{255}, \epsilon_2 = \frac{36}{255}$) experiments.

| Methods | Runtime (s) |
|---|---|
| $l_\infty$ | 182 |
| $l_2$ | 165 |
| CURE-Joint | 320 |
| CURE-Max | 155 |
| CURE-Random | 190 |
| CURE-Finetune | 148 |
| CURE-Scratch | 154 |

Table 6: Runtime for all methods on MNIST ($\epsilon_\infty = 0.1, \epsilon_2 = 0.5$) experiment per epoch in seconds.

**Runtime for CURE-Scratch on MNIST, CIFAR10, and TinyImagenet datasets.** In Table 7, we show the runtime per training epoch using **CURE-Scratch** on MNIST, CIFAR10, and TinyImagenet datasets with and without GP operations. We see that the GP operation's cost is small compared with the whole training procedure, accounting for around 6% of the whole training time.

| | MNIST | CIFAR-10 | TinyImagenet |
|---|---|---|---|
| w/o GP | 148 | 390 | 952 |
| with GP | 154 | 414 | 1036 |

Table 7: Runtime for CURE-Scratch on MNIST, CIFAR10, and TinyImagenet datasets.

# D ALGORITHMS

In this section, we present the algorithms of **CURE** framework. Algorithm 1 illustrates how to get propagation region for both $l_2$ and $l_\infty$ perturbations. Algorithm 2, 3, 4, 5 refer to algorithms of CURE-Joint, CURE-Max, CURE-Random, and CURE-Scratch/Finetune, respectively. Algorithm 6

is the procedure of performing GP after one epoch of natural and certified training (could be any of Algorithm 2, 3, 4, 5).

---

**Algorithm 1** get_propagation_region for $l_\infty$ and $l_2$ perturbations

---

**Input:** Neural network $f$, input $\boldsymbol{x}$, label $t$, perturbation radius $\epsilon$, subselection ratio $\lambda$, step size $\alpha$, step number $n$, attack types $\in \{l_\infty, l_2\}$

**Output:** Center $\boldsymbol{x}'$ and radius $\tau$ of propagation region $\mathcal{B}^\tau(\boldsymbol{x}')$

    $(\underline{\boldsymbol{x}}, \overline{\boldsymbol{x}}) \leftarrow \text{clamp}((\boldsymbol{x} - \epsilon, \boldsymbol{x} + \epsilon), 0, 1)$          // Get bounds of input region

    $\boldsymbol{\tau} \leftarrow \lambda/2 \cdot (\overline{\boldsymbol{x}} - \underline{\boldsymbol{x}})$               // Compute propagation region size $\tau$

    $\boldsymbol{x}_0^* \leftarrow \text{Uniform}(\underline{\boldsymbol{x}}, \overline{\boldsymbol{x}})$             // Sample PGD initialization

    **for** $i = 0 \ldots n - 1$ **do**             // Do $n$ PGD steps

        **if** attack = $l_\infty$ **then**          // PGD-$l_\infty$

            $\boldsymbol{x}_{i+1}^* \leftarrow \boldsymbol{x}_i^* + \alpha \cdot \epsilon \cdot \text{sign}(\nabla_{\boldsymbol{x}_i^*} \mathcal{L}_{\text{CE}}(f(\boldsymbol{x}_i^*), t))$

            $\boldsymbol{x}_{i+1}^* \leftarrow \text{clamp}(\boldsymbol{x}_{i+1}^*, \underline{\boldsymbol{x}}, \overline{\boldsymbol{x}})$

        **end if**

        **if** attack = $l_2$ **then**            // PGD-$l_2$

            $\boldsymbol{x}_{i+1}^* \leftarrow \boldsymbol{x}_i^* + \alpha \cdot \frac{\nabla_{\boldsymbol{x}_i^*} \mathcal{L}_{\text{CE}}(f(\boldsymbol{x}_i^*), \boldsymbol{y})}{\|\nabla_{\boldsymbol{x}_i^*} \mathcal{L}_{\text{CE}}(f(\boldsymbol{x}_i^*), \boldsymbol{y})\|_2}$

            $\delta \leftarrow \frac{\epsilon}{\|\boldsymbol{x}_{i+1}^* - \boldsymbol{x}\|_2} \cdot (\boldsymbol{x}_{i+1}^* - \boldsymbol{x})$

            $\boldsymbol{x}_{i+1}^* \leftarrow \text{clamp}(\boldsymbol{x} + \delta, \underline{\boldsymbol{x}}, \overline{\boldsymbol{x}})$

        **end if**

    **end for**

    $\boldsymbol{x}' \leftarrow \text{clamp}(\boldsymbol{x}_n^*, \underline{\boldsymbol{x}} + \tau, \overline{\boldsymbol{x}} - \tau)$        // Ensure that $\mathcal{B}^\tau(\boldsymbol{x}')$ will lie fully in $\mathcal{B}^\epsilon(\boldsymbol{x})$

    **return** $\boldsymbol{x}', \tau$

---

**Algorithm 2** CURE-Joint Training Epoch

---

**Input:** Neural network $f_\theta$, training set $(\boldsymbol{X}, \boldsymbol{T})$, perturbation radius $\epsilon_2$ and $\epsilon_\infty$, subselection ratio $\lambda_\infty$ and $\lambda_2$, learning rate $\eta$, $\ell_1$ regularization weight $\ell_1$, loss balance factor $\alpha$

    **for** $(\boldsymbol{x}, t) = (\boldsymbol{x}_0, t_0) \ldots (\boldsymbol{x}_b, t_b)$ **do**          // Sample batches $\sim (\boldsymbol{X}, \boldsymbol{T})$

        $(\boldsymbol{x}_\infty', \tau_\infty) \leftarrow \text{get\_propagation\_region} \, (\text{attack} = l_\infty)$ // Refer to Algorithm 1

        $(\boldsymbol{x}_2', \tau_2) \leftarrow \text{get\_propagation\_region} \, (\text{attack} = l_2)$

        $\mathcal{B}^{\tau_\infty}(\boldsymbol{x}_\infty') \leftarrow \text{BOX}(\boldsymbol{x}_\infty', \tau_\infty)$        // Get box with midpoint $\boldsymbol{x}_\infty', \boldsymbol{x}_2'$ and radius $\tau_\infty, \tau_2$

        $\mathcal{B}^{\tau_2}(\boldsymbol{x}_2') \leftarrow \text{BOX}(\boldsymbol{x}_2', \tau_2)$

        $\boldsymbol{u}_{y_\infty^\triangle} \leftarrow \text{get\_upper\_bound}(f_\theta, \mathcal{B}^{\tau_\infty}(\boldsymbol{x}_\infty'))$    // Get upper bound $\boldsymbol{u}_{y_\infty^\triangle}, \boldsymbol{u}_{y_2^\triangle}$ on logit differences

        $\boldsymbol{u}_{y_2^\triangle} \leftarrow \text{get\_upper\_bound}(f_\theta, \mathcal{B}^{\tau_2}(\boldsymbol{x}_2'))$      // based on IBP

        $\text{loss}_{l_\infty} \leftarrow \mathcal{L}_{\text{CE}}(\boldsymbol{u}_{y_\infty^\triangle}, t)$

        $\text{loss}_{l_2} \leftarrow \mathcal{L}_{\text{CE}}(\boldsymbol{u}_{y_2^\triangle}, t)$

        $\text{loss}_{\ell_1} \leftarrow \ell_1 \cdot \text{get\_}\ell_1\text{\_norm}(f_\theta)$

        $\text{loss}_{tot} \leftarrow (1 - \alpha) \cdot \text{loss}_{l_\infty} + \alpha \cdot \text{loss}_{l_2} + \text{loss}_{\ell_1}$

        $\theta \leftarrow \theta - \eta \cdot \nabla_\theta \text{loss}_{tot}$          // Update model parameters $\theta$

    **end for**

---

---

**Algorithm 3** CURE-Max Training Epoch

---

**Input:** Neural network $f_\theta$, training set $(\boldsymbol{X}, \boldsymbol{T})$, perturbation radius $\epsilon_2$ and $\epsilon_\infty$, subselection ratio $\lambda_\infty$ and $\lambda_2$, learning rate $\eta$, $\ell_1$ regularization weight $\ell_1$

 **for** $(\boldsymbol{x}, t) = (\boldsymbol{x}_0, t_0) \dots (\boldsymbol{x}_b, t_b)$ **do**     // Sample batches $\sim (\boldsymbol{X}, \boldsymbol{T})$
  $(\boldsymbol{x}'_\infty, \tau_\infty) \leftarrow$ get_propagation_region (attack $= l_\infty$) // Refer to Algorithm 1
  $(\boldsymbol{x}'_2, \tau_2) \leftarrow$ get_propagation_region (attack $= l_2$)
  $\mathcal{B}^{\tau_\infty}(\boldsymbol{x}'_\infty) \leftarrow \text{Box}(\boldsymbol{x}'_\infty, \tau_\infty)$    // Get box with midpoint $\boldsymbol{x}'_\infty, \boldsymbol{x}'_2$ and radius $\tau_\infty, \tau_2$
  $\mathcal{B}^{\tau_2}(\boldsymbol{x}'_2) \leftarrow \text{Box}(\boldsymbol{x}'_2, \tau_2)$
  $\boldsymbol{u}_{y^\Delta_\infty} \leftarrow$ get_upper_bound$(f_\theta, \mathcal{B}^{\tau_\infty}(\boldsymbol{x}'_\infty))$  // Get upper bound $\boldsymbol{u}_{y^\Delta_\infty}, \boldsymbol{u}_{y^\Delta_2}$ on logit differences
  $\boldsymbol{u}_{y^\Delta_2} \leftarrow$ get_upper_bound$(f_\theta, \mathcal{B}^{\tau_2}(\boldsymbol{x}'_2))$  // based on IBP
  $\text{loss}_{l_\infty} \leftarrow \mathcal{L}_{\text{CE}}(\boldsymbol{u}_{y^\Delta_\infty}, t)$
  $\text{loss}_{l_2} \leftarrow \mathcal{L}_{\text{CE}}(\boldsymbol{u}_{y^\Delta_2}, t)$
  $\text{loss}_{Max} \leftarrow max(\text{loss}_{l_\infty}, \text{loss}_{l_2})$   // We select the largest $l_{p \in [2,\infty]}$ loss for each sample
  $\text{loss}_{\ell_1} \leftarrow \ell_1 \cdot$ get_$\ell_1$_norm$(f_\theta)$
  $\text{loss}_{tot} \leftarrow \text{loss}_{Max} + \text{loss}_{\ell_1}$
  $\theta \leftarrow \theta - \eta \cdot \nabla_\theta \text{loss}_{tot}$      // Update model parameters $\theta$
 **end for**

---

**Algorithm 4** CURE-Random Training Epoch

---

**Input:** Neural network $f_\theta$, training set $(\boldsymbol{X}, \boldsymbol{T})$, perturbation radius $\epsilon_2$ and $\epsilon_\infty$, subselection ratio $\lambda_\infty$ and $\lambda_2$, learning rate $\eta$, $\ell_1$ regularization weight $\ell_1$

 **for** $(\boldsymbol{x}, t) = (\boldsymbol{x}_0, t_0) \dots (\boldsymbol{x}_b, t_b)$ **do**     // Sample batches $\sim (\boldsymbol{X}, \boldsymbol{T})$
  $(\boldsymbol{x}_1, \boldsymbol{x}_2), (t_1, t_2) \leftarrow$ partition$(\boldsymbol{x}, t)$   // Randomly partition inputs into two blocks
                // Apply Algorithm 1
  $(\boldsymbol{x}'_\infty, \tau_\infty) \leftarrow$ get_propagation_region $(\boldsymbol{x}_1, t_1, \text{attack} = l_\infty)$
  $(\boldsymbol{x}'_2, \tau_2) \leftarrow$ get_propagation_region $(\boldsymbol{x}_2, t_2, \text{attack} = l_2)$
  $\mathcal{B}^{\tau_\infty}(\boldsymbol{x}'_\infty) \leftarrow \text{Box}(\boldsymbol{x}'_\infty, \tau_\infty)$    // Get box with midpoint $\boldsymbol{x}'_\infty, \boldsymbol{x}'_2$ and radius $\tau_\infty, \tau_2$
  $\mathcal{B}^{\tau_2}(\boldsymbol{x}'_2) \leftarrow \text{Box}(\boldsymbol{x}'_2, \tau_2)$
  $\boldsymbol{u}_{y^\Delta_\infty} \leftarrow$ get_upper_bound$(f_\theta, \mathcal{B}^{\tau_\infty}(\boldsymbol{x}'_\infty))$  // Get upper bound $\boldsymbol{u}_{y^\Delta_\infty}, \boldsymbol{u}_{y^\Delta_2}$ on logit differences
  $\boldsymbol{u}_{y^\Delta_2} \leftarrow$ get_upper_bound$(f_\theta, \mathcal{B}^{\tau_2}(\boldsymbol{x}'_2))$  // based on IBP
  $\text{loss}_{l_\infty} \leftarrow \mathcal{L}_{\text{CE}}(\boldsymbol{u}_{y^\Delta_\infty}, t)$
  $\text{loss}_{l_2} \leftarrow \mathcal{L}_{\text{CE}}(\boldsymbol{u}_{y^\Delta_2}, t)$
  $\text{loss}_{\ell_1} \leftarrow \ell_1 \cdot$ get_$\ell_1$_norm$(f_\theta)$
  $\text{loss}_{tot} \leftarrow \text{loss}_{l_\infty} + \text{loss}_{l_2} + \text{loss}_{\ell_1}$
  $\theta \leftarrow \theta - \eta \cdot \nabla_\theta \text{loss}_{tot}$      // Update model parameters $\theta$
 **end for**

---

---

**Algorithm 5** CURE-Scratch/Finetune Training Epoch

---

**Input:** Neural network $f_\theta$, training set $(\boldsymbol{X}, \boldsymbol{T})$, perturbation radius $\epsilon_2$ and $\epsilon_\infty$, subselection ratio $\lambda_\infty$ and $\lambda_2$, learning rate $\eta$, $\ell_1$ regularization weight $\ell_1$, KL loss balance factor $\eta$, mode $\in$ [Scratch, Finetune]

$\quad$**for** $(\boldsymbol{x}, t) = (\boldsymbol{x}_0, t_0) \ldots (\boldsymbol{x}_b, t_b)$ **do** $\qquad$ // Sample batches $\sim (\boldsymbol{X}, \boldsymbol{T})$

$\quad\quad (\boldsymbol{x}'_\infty, \tau_\infty) \leftarrow$ get_propagation_region $(\text{attack} = l_\infty)$ // Refer to Algorithm 1

$\quad\quad (\boldsymbol{x}'_2, \tau_2) \leftarrow$ get_propagation_region $(\text{attack} = l_2)$

$\quad\quad \mathcal{B}^{\tau_\infty}(\boldsymbol{x}'_\infty) \leftarrow \text{BOX}(\boldsymbol{x}'_\infty, \tau_\infty)$ $\qquad$ // Get box with midpoint $\boldsymbol{x}'_\infty, \boldsymbol{x}'_2$ and radius $\tau_\infty, \tau_2$

$\quad\quad \mathcal{B}^{\tau_2}(\boldsymbol{x}'_2) \leftarrow \text{BOX}(\boldsymbol{x}'_2, \tau_2)$

$\quad\quad \boldsymbol{u}_{y_\infty^\Delta} \leftarrow$ get_upper_bound$(f_\theta, \mathcal{B}^{\tau_\infty}(\boldsymbol{x}'_\infty))$ $\quad$ // Get upper bound $\boldsymbol{u}_{y_\infty^\Delta}, \boldsymbol{u}_{y_2^\Delta}$ on logit differences

$\quad\quad \boldsymbol{u}_{y_2^\Delta} \leftarrow$ get_upper_bound$(f_\theta, \mathcal{B}^{\tau_2}(\boldsymbol{x}'_2))$ $\qquad$ // based on IBP

$\quad\quad \text{loss}_{l_\infty} \leftarrow \mathcal{L}_{\text{CE}}(\boldsymbol{u}_{y_\infty^\Delta}, t)$

$\quad\quad \text{loss}_{l_2} \leftarrow \mathcal{L}_{\text{CE}}(\boldsymbol{u}_{y_2^\Delta}, t)$

$\quad\quad \text{loss}_{Max} \leftarrow max(\text{loss}_{l_\infty}, \text{loss}_{l_2})$ $\qquad$ // We select the largest $l_{p \in [2,\infty]}$ loss for each sample

$\quad\quad \text{loss}_{\ell_1} \leftarrow \ell_1 \cdot$ get_$\ell_1$_norm$(f_\theta)$

$\quad\quad$ find correctly certified $l_q$ subset $\gamma$ using Definition 4.2

$\quad\quad \text{loss}_{KL} \leftarrow KL(d_q[\gamma] \| d_r[\gamma])$ $\qquad$ // Eq. 2

$\quad\quad \text{loss}_{tot} \leftarrow \text{loss}_{Max} + \eta \cdot \text{loss}_{KL} + \text{loss}_{\ell_1}$

$\quad\quad \theta \leftarrow \theta - \eta \cdot \nabla_\theta \text{loss}_{tot}$ $\qquad$ // Update model parameters $\theta$

$\quad$**end for**

---

**Algorithm 6** GP: Connect CT with NT

---

1: **Input**: model $f_\theta$, input images with distribution $\mathcal{D}$, training rounds $R$, $\beta$, natural training **NT** and certified training **CT** algorithms, perturbation radius $\epsilon_\infty$ and $\epsilon_2$, subselection ratio $\lambda_\infty$ and $\lambda_2$, learning rate $\eta$, $\ell_1$ regularization weight $\ell_1$.

2:

3: **for** $r = 1, 2, ..., R$ **do**

4: $\quad f_n \leftarrow \textbf{NT}(f_\theta^{(r)}, \mathcal{D})$

5: $\quad f_c \leftarrow \textbf{CT}(f_\theta^{(r)}, \epsilon_\infty, \epsilon_2, \lambda_\infty, \lambda_2, \eta, \ell_1, \mathcal{D})$ $\qquad$ // Can be single-norm or any CURE training

6: $\quad$ compute $g_n \leftarrow f_n - f_\theta^{(r)}$, $g_c \leftarrow f_c - f_\theta^{(r)}$

7: $\quad$ compute $g_p$ using Eq. 7

8: $\quad$ update $f_\theta^{(r+1)}$ using Eq. 8 with $\beta$ and $g_c$

9: **end for**

10: **Output**: model $f_\theta$.

---

