# OpenReview forum: "Towards Universal Certified Robustness with Multi-Norm Training"
_ICLR.cc/2025/Conference — ICLR 2025 Conference Withdrawn Submission_

### Official Review · Reviewer_Ve9E · 2024-10-24

**Soundness:** 3
**Presentation:** 3
**Contribution:** 3
**Rating:** 6
**Confidence:** 5

**Summary:**

The paper explores the connection between certified training with multiple perturbation regions. The authors propose a framework for joint training with multiple variants and incremental improvements. The proposed method improves the union certified accuracy and also generalizes to certified robustness with respect to different geometric transformations that were not specifically used during training.

**Strengths:**

The paper addresses a challenge that was (to the best of my knowledge) not addressed before. The proposed multi-norm certified training methods improve performance over previous SOTA methods that don’t use multi-norm certified training. Training certifiably robust networks jointly for multiple perturbation types seems like a promising approach toward universally robust models.

The paper is well-written and easy to follow. There are plenty of experimental results, and the insights gained from this work are valuable.

**Weaknesses:**

* Even if $L_2$ and $L_\infty$ perturbations are generally studied in this field, given the claim of universality, it would be interesting to have some discussion about other perturbation types (certification and/or training attempts).
* The $L_2$ certified training method proposed suffers from a significant drop in natural accuracy that can only be mitigated by using a very small value for $\lambda_2$, which becomes very close to just using Adversarial Training instead. I would like to see some results obtained by using AT (equivalent to $\lambda_2 = 0$). I suspect the main cause for this to be the complex interaction between the propagated $L_\infty$ box as a proxy for $L_2$ robustness. Consider this simple case: we have a data point $x$ which is a $n^2$ dimensional point ($n=28$ for MNIST, $n=32 \sqrt 3$ for CIFAR). We want to train for robustness in a $L_2$ ball with radius \epsilon, using subselection rate $\lambda$. The algorithm searches for an adversarial example ($x+\delta$) inside the $L_2$ ball with radius $\epsilon(1-\lambda) = \epsilon - \tau$ and then propagates the $L_\infty$ ball with radius $\tau$ centered in $x+\delta$. Consider the case when $\delta = 1 \cdot (\epsilon-\tau)/n$ (all values of $\delta$ are the same and $||\delta|| =\epsilon-\tau$ ). Let $\beta = 1 \cdot \tau$. Then the point $\tilde x =  x + \delta + \beta$ is the most distant point from $x$ that is contained within the $L_\infty$ ball that is being propagated. The $L_2$ distance between $\tilde x$ and $x$ is $||\delta + \beta|| = \epsilon - \tau + n \tau = \epsilon + (n-1) \tau = \epsilon (1 + (n-1) \lambda)$. For a seemingly normal value of $\lambda = 0.1$ (the value used by SABR for CIFAR-10 $\epsilon=2/255$) this distance can be $4$ to $6$ times larger than $\epsilon$, which will introduce a lot of unnecessary regularization, explaining why you had to resort to much smaller values for $\lambda_2$. It would be interesting to analyze some other options/strategies for constructing the $L_\infty$ box in this case as it is very likely that this would further improve the results.
* The Bound Alignment procedure seems complex and not perfectly motivated. It is not clear why we want to align the bounds for the two perturbations. It is also not very clear if the alignment of the bounds is computed based on the small Box obtained after the adversarial attack during training. Definition 4.2 seems to imply that you use the bounds for the whole eps-Ball, but Algorithm 5 shows that loss_KL is computed for the bounds of the small box.

**Minor points**
* For reproducibility it would be nice to have a paragraph or table with all hyperparameters needed for experiments. For example, there are no given values for the $\lambda_\infty$ subselection rate. Instead, they are just cited to be the same as SABR.
* It would be nice to have a short description of how the certification methods used for evaluation work in different scenarios as readers might not be familiar with all of them.
* Figures about bound alignment (3,5,6,7,8,9) are not very expressive and it is not easy to draw conclusions based on them. Also, I don’t understand what should be the difference between Fig 8 and 9. I think they show different test samples, but this should be more clearly stated in the text.

**Questions:**

1. Is it possible to include geometric perturbations in the training framework?
2. The results of CURE on MNIST + Translation transformation are unexpectedly low. Do you have any insights on why this happens? Do you also have results for CIFAR-10?
3. It is not very clear when GP is applied during training. Appendix A states *We only apply GP after training with the final epsilon values*. Does this mean GP is applied after the eps-annealing phase?
4. Would it be possible to obtain a similar effect to Bound Alignment if we just give samples from $\gamma$ a higher weight for the $L_r$ loss?
5. The learning rate is reported as 1e-4, but SABR uses 5e-4. Is this a writing mistake or a deliberate change?

---

### Official Review · Reviewer_ZgAU · 2024-10-28

**Soundness:** 2
**Presentation:** 3
**Contribution:** 1
**Rating:** 3
**Confidence:** 4

**Summary:**

This paper introduces CURE, a certified training framework that advances the state-of-the-art in multi-norm robustness certification. The framework's key innovation is twofold: it presents the deterministic l2-norm certified training defense and develops comprehensive multi-norm certified training methodologies.

**Strengths:**

This work is well-written and easy to understand. The certified robustness issue is significant, but it would benefit substantially from more rigorous formal analyses of certified robustness.

**Weaknesses:**

This work is not self-consistant, e.g., it lacks technical details regarding the implementation of the IBP loss computation, particularly in deriving the lower and upper bounds.

The work appears to be a straightforward adaptation of multi-norm adversarial training ideas to certified training, without providing deeper (theoretical) insights. A rigorous analysis explaining why this framework is specifically effective for certified training would significantly strengthen the contribution. The current version seems not interesting. Could authors provide a theoretical justification for why this approach can improve certified robustness?

The paper notably lacks computational complexity analysis. Given the typically high computational demands of certified training methods, this omission is particularly significant.

The experiments could be more comprehensive, particularly in comparison with recent advances in adversarial training with DDPM-generated data. ref https://github.com/wzekai99/DM-Improves-AT.  I guess DDPM-data can also help to improve certified robustness.

Rough definition: Could authors provide formal definitions in 4.1 and 4.2?

The logical relationship in lines 282-284 requires clarification regarding necessary or sufficient conditions. E.g., in the worst case, A_u = A_q = 0, does it mean optimizing A_u->A_q will reach the worst?

In my opinion, the most ideal case is A_u = A_q = 100\%. Why optimizing A_u->A_q is better than optimizing A_u -> 100\%?

Given a specific training set, I guess the ideal case may not satisfy A_u = A_q. Could authors prove that the ideal case must satisfy A_u = A_q given any training set drawn from an unknown data distribution?

Even though a model is optimized to A_u = A_q, how to prove it is the best over the current training set?

Line 297-298: Since A_q serves as the upper bound of A_u, similarly, $\gamma$ can be regarded as... Why $\gamma$ can be??? Please justify "more likely". I cannot find the logic/causality here.

**Questions:**

Ref Weaknesses.

The current manuscript does not meet ICLR's bar for several fundamental reasons. The core contribution—balancing losses across different norms in certified training—appears incremental without substantial theoretical innovation. Critical questions remain unaddressed: the paper lacks theoretical justification for why this multi-norm framework is specifically effective in certified training, as opposed to general adversarial training; there are so many "logical" problems in Sec. 4.2.  Moreover, there is insufficient analysis demonstrating how the proposed method theoretically guarantees improvements in certified robustness. The absence of formal analysis of certified robustness significantly weakens the paper's contribution to the field.

---

### Official Review · Reviewer_QTcj · 2024-11-06

**Soundness:** 2
**Presentation:** 2
**Contribution:** 2
**Rating:** 3
**Confidence:** 4

**Summary:**

The paper introduces the CURE framework -- a selection of related methods for training certifiably robust models. The introduced methods are derived from SABR (Muller et al.).

The paper proposes gradient projection and bound alignment techniques for improving certified robustness against multi-norm norm bounded perturbations as well as geometric perturbations -- while conducting evaluations on vision tasks (CIFAR-10, MNIST and TinyImagenet) with small networks.

**Strengths:**

The proposed method shows that using gradient propagation and bound alignment can result in good performance wins on multi-norm certified robustness as well as on geometric perturbations.

Plenty of evaluations of multi norm certified accuracy, on L-inf, L-2 and geometric transformations -- on a few vision datasets with documented settings.

All hyper-parameters are given in the appendix and pseudo-code, which should make the method more reproducible.

**Weaknesses:**

The main issue is the scope and clarity of contributions of the paper.

One of the main contributions is claimed to be a method for L2 certified training. The paper is replacing a (hyper-)box with a (hyper-)sphere in the SABR method, without any analysis or theoretical justification. This is a baseline, not a new methods.

The three variants of CURE: Joint, Max, Random are also claimed contributions. These methods just naturally arise from having different methods for dealing with L2 and L-inf norm bounded perturbations. So, they also naturally seem to be baselines.

CURE-Finetune applies CURE-Scratch on a different base model (i.e. a pre-trained one) with some other small changes. It does not seem to be a different method, so there is little need for a new name.

Overall, it is a bit confusing that "CURE" is a "framework". It seems that the paper would be much clearer if it would separate out the actual contributions and focus on them: bound alignment, gradient projection and changes needed for fine-tuning -- and disentangle/isolate and ablate all those components individually (table 3 seems to do this, at a very small scale).


Other issues include:

- Very confusing notation overall. For example, what is "A_u \rightarrow A_q" in line 283. Also, equation 2 looks very clumsy (could the indices be subscripts/superscripts/etc.?). Many equations do not seem to be needed at all; equation 7 takes a union of parameters and 8 shows a linear combination; etc.

- Irrelevant definitions (4.1 and 4.2) and an extreme amount of mathematical notation for concepts that could really be explained much more clearly with just words (all of section 4 could be written in half the space). Note that there is no theoretical justification given, not proofs, bounds, analysis, etc.

- Confusing terminology: "certified update", line 329; "certifiably fine-tuning", line 353; etc.

The paper definitely has some good parts, but in the current state, without a major re-write, I cannot recommend acceptance.

**Questions:**

See above.

---

### Official Review · Reviewer_hXCa · 2024-11-07

**Soundness:** 2
**Presentation:** 2
**Contribution:** 2
**Rating:** 3
**Confidence:** 4

**Summary:**

Current certified training methods only make models robust to specific perturbation types, but fail to provide robustness across multiple perturbation types simultaneously. The authors propose CURE, a novel multi-norm certified training framework that enhances robustness across different perturbations by combining deterministic defenses with multi-norm certified training techniques. CURE achieves up to 23.9% improvement in union robustness and better generalization on unseen transformations, marking a step toward universal certified robustness.

**Strengths:**

1.This paper showed a new unified adversarial training method for certified robustness.

2.The experimental results demonstrates the effiectiveness of the proposed method.

**Weaknesses:**

1.The new loss proposed in this paper appears to be merely an adaptation of the unified loss used in general adversarial training, which significantly detracts from the originality and contribution of this work.

2.The unified loss considers only the l2 and l infty cases, which, from the reviewer's perspective, falls short of guaranteeing universal certified robustness. The authors should include results across a wider range of lp norms to substantiate this claim.

3.The authors do not provide a theoretical analysis demonstrating how the proposed loss function contributes to improved universal certified robustness.

**Questions:**

N/A

---

### Note · Authors · 2024-11-18

I have read and agree with the venue's withdrawal policy on behalf of myself and my co-authors.